# Contrasting Effects of Chinese Fir Plantations of Different Stand Ages on Soil Enzyme Activities and Microbial Communities

**Chaoqun Wang** [1,2,3], **Lin Xue** [2,3,4], **Yuhong Dong** [1,2,3], **Lingyu Hou** [1,2,3], **Yihui Wei** [1,2,3], **Jiaqi Chen** [1,2,3] **and Ruzhen Jiao** [1,2,3,*]

1   Research Institute of Forestry, Chinese Academy of Forestry, Beijing 100091, China;
    wcq930207@163.com (C.W.); dongyh@caf.ac.cn (Y.D.); houlingyu@caf.ac.cn (L.H.);
    m13240341188@163.com (Y.W.); 18562316708@163.com (J.C.)
2   State Key Laboratory of Tree Genetics and Breeding, Chinese Academy of Forestry, Beijing 100091, China;
    xuelin@caf.ac.cn
3   Key Laboratory of Tree Breeding and Cultivation of State Forestry Administration, Chinese Academy of
    Forestry, Beijing 100091, China
4   Research Institute of Forest Ecology, Environment and Protection, Chinese Academy of Forestry,
    Beijing 100091, China
*   Correspondence: jiaorzh@caf.ac.cn; Tel.: +86-010-628-89663

**Abstract:** Soil enzymes and microbial communities are key factors in forest soil ecosystem functions and are affected by stand age. In this study, we studied soil enzyme activities, composition and diversity of bacterial and fungal communities and relevant physicochemical properties at 0–10 cm depth (D1), 10–20 cm depth (D2) and 20–30 cm depth (D3) soil layers in 3-(3a), 6-(6a), 12-(12a), 18-(18a), 25-(25a), 32-(32a) and 49-year-old (49a) Chinese fir plantations to further reveal the effects of stand age on soil biotic properties. Spectrophotometry and high-throughput sequencing was used to assess the soil enzyme activity and microbial community composition and diversity of Chinese fir plantation of different stand ages, respectively. We found that soil catalase activity increased as the stand age of Chinese fir plantations increased, whereas the activities of urease, sucrase and β-glucosidase in 12a, 18a and 25a were lower than those in 6a, 32a and 49a. Shannon and Chao1 indices of bacterial and fungal communities first decreased gradually from 6a to 18a or 25a and then increased gradually from 25a to 49a. Interestingly, the sucrase and β-glucosidase activities and the Shannon and Chao1 indices in 3a were all lower than 6a. We found that the relative abundance of dominant microbial phyla differed among stand ages and soil depths. The proportion of Acidobacteria first increased and then decreased from low forest age to high forest age, and its relative abundance in 12a, 18a and 25a were higher than 3a, 32a and 49a, but the proportion of Proteobacteria was opposite. The proportion of Ascomycota first decreased and then increased from 6a to 49a, and its relative abundance in 12a, 18a and 25a was lower than 3a, 6a, 32a and 49a. Our results indicate that soil enzyme activities and the richness and diversity of the microbial community are limited in the middle stand age (from 12a to 25a), which is important for developing forest management strategies to mitigate the impacts of degradation of soil biological activities.

**Keywords:** stand age; bacterial community; fungal community; enzymes

## 1. Introduction

Plant–microbe interactions are central to soil fertility and ecosystem function. In both natural and managed ecosystems, research has demonstrated the strong selective effects of plants on soil microbial

communities and how resultant soil microbial communities may directly impact plant communities and soil nutrient cycling [1–3]. Soil enzymes and microorganisms are useful biotic indicators of soil fertility and ecosystem functions because they are the main drivers of soil nutrient cycling and carbon cycling [4]. Catalase (CAT) is a specific enzyme for the metabolism of hydrogen peroxide [5]; sucrase (SC) plays an important role in decomposing sucrose and promoting the conversion of activated carbon [6]; β-glucosidase (BG) is mainly involved in the degradation of cellulose [4]; urease (UE) mainly promotes the mineralization of nitrogen (N) [7]. Microorganisms play an important role in the production of enzymes, and the production of specific enzymes can be partly explained by the need of microorganisms to limit nutrients [8].

In general, studies of ecosystem succession have been central in demonstrating that forest ecosystems of increasing age can impact microbial community structure in variable ways over time. For example, there can be increasing selective effects of plant communities on microbial community structure and related function [9,10]. As microbial communities mediate biogeochemical cycles associated with nutrient and carbon cycling, in managed forest ecosystems the influence of increasing stand age on microbial communities may have strong effects on ecosystem functioning and soil fertility via soil enzyme activity [11,12]. The results of a study on forests in the Mid-Atlantic US demonstrated that the concentrations of Mg, Ca, $NO_3$ and pH in young forest were higher than those in old forest [13]. Zhang et al. [14] reported that soil nutrient contents and microbial biomass first gradually increased from <1a to 9a and then gradually decreased with the increase of stand age in *Lycium barbarum* L. plantation. Li et al. [15] found that certain correlations between soil microbial community diversity and stand age of seabuckthorn forest were obvious. Luo et al. [16] reported that soil enzyme activities, such as β-glucosidase, urease and protease, gradually increase with increasing stand age of Seabuckthorn (*Hippophae rhamnoides* Linn.) plantation. Therefore, determining the response of the microbial community and enzyme activity to changes in stand age can provide a theoretical basis for developing forest management strategies.

Several studies have reported that stand age can drive changes in soil properties of Chinese fir plantations [17–19]. Nevertheless, the influence of stand age on soil enzyme activities and microbial diversity has not been agreed upon, and the effect of stand age on the soil microbial community composition of Chinese fir plantations is still unclear. Wu et al. [19] reported that β-glucosidase activity first decreased and then increased with the increase of stand age, while invertase and polyphenol oxidase activities were the opposite. Shu [20] found that the activities of acid phosphatase, catalase, urease and protease gradually decreased as the forest aged. Wang et al. [21] found that the number of microorganisms first decreased and then increased with the increase of stand age, while Liu et al. [22] reported that the microbial diversity generally increased with the increase of stand age of Chinese fir plantation. Therefore, it is necessary to further study the soil enzyme activities, microbial community composition and diversity of Chinese fir plantations of different stand ages. In this study, we proposed three hypotheses: (1) stand age will impact soil enzyme activity; (2) stand will impact soil microbial composition and diversity; and (3) the effects of stand age on enzyme activity and soil microbes are not completely positive or negative.

## 2. Materials and Methods

### 2.1. Site Selection and Soil Sampling

Soil samples were taken from the Shanxia Forest Farm in late December 2017. We first divided the forest map into three blocks. Each block consisted of seven stand age plantations (3-(3a), 6-(6a), 12-(12a), 18-(18a), 25-(25a), 32-(32a), and 49-year-old (49a) Chinese fir plantations that are located as close to each other as possible (Figure 1). Each plantation was given three plots within each block, each of which was 20m by 30m, in a total of 63 plots. The management measures for all plantations are consistent. All plantations were all burned before planting, and were not thinned and were not fertilized during stand growth. Soil samples were collected at three depths (D1: 0–10 cm, D2: 10–20 cm

and D3: 20–30 cm). Differences in the sampling plots of different ages are shown in Table 1. Five soil cores were randomly sampled from each plot. At the same time, soil from three depths was collected using a ring cutter with 100 cm$^3$ volumes for measuring soil bulk density. Soil samples for analyses were immediately sieved (2 mm) and were stored at 4 °C. After being transported to the laboratory, soil samples stored at −80 °C prior to microbial community analysis, or air dried and ground prior to chemical properties analysis.

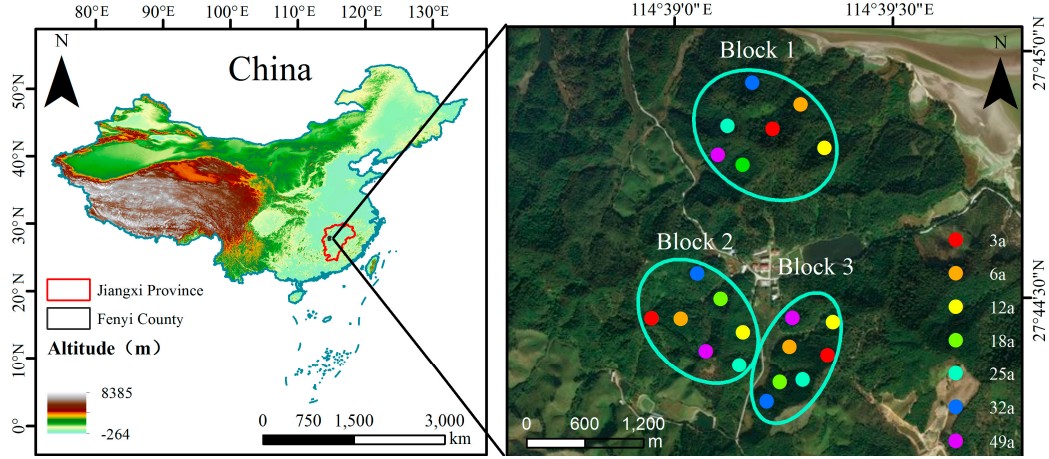

**Figure 1.** Locations of sampling blocks.

**Table 1.** Characteristics of the plots of different ages.

| Stand Age (a) | Mean Elevation (m) | Slope (°) | Aspects | Existing Density (stems·hm$^{-2}$) | Dominant Undergrowth Vegetation Species |
|---|---|---|---|---|---|
| 3a | 106–113 | 26–27 | southeast | 3333 | *Blechnum japonica*, *Rubus trianthus* Foeke., *Lophatherum gracile*, *Ophiopogon japonicus*, *Maesa japonica* (Thunb) Moritzi, *Efigeron acris* L., *Adinandra milletti* (Hook. Et Arn.) Benth, *Clerodendrum cyrtophyllum* Turcz. |
| 6a | 106–116 | 25–27 | southeast, south | 3333 | *Blechnum japonica*, *Maesa japonica* (Thunb) Moritzi, *Clerodendrum cyrtophyllum* Turcz. |
| 12a | 106–107 | 28–32 | southeast | 3294 | *Blechnum japonica*, *Maesa japonica* (Thunb) Moritzi |
| 18a | 105–132 | 20–23 | southeast | 3067 | *Blechnum japonica*, *Rubus trianthus* Foeke., *Maesa japonica* (Thunb) Moritzi, *Efigeron acris* L. |
| 25a | 93–122 | 25–30 | Southeast, south | 2589 | *Blechnum japonica*, *Rubus trianthus* Foeke., *Lophatherum gracile*, *Maesa japonica* (Thunb) Moritzi, *Efigeron acris* L., *Camellia oleifera* Abel. |
| 32a | 103–138 | 28–32 | Southeast, south | 2161 | *Blechnum japonica*, *Rubus trianthus* Foeke., *Smilax glabra* Roxb., *Maesa japonica* (Thunb) Moritzi, *Efigeron acris* L., *Camellia oleifera* Abel. |
| 49a | 93–139 | 25–30 | Southeast, south | 1968 | *Blechnum japonica*, *Rubus trianthus* Foeke., *Maesa japonica* (Thunb) Moritzi, *Efigeron acris* L., *Camellia oleifera* Abel., *Schima superba* Garde. Et Champ. |

## 2.2. Soil Physicochemical Analyses

Soil bulk density (SBD) and alkali hydrolysis nitrogen (AN) content was determined according to literature [23]. Soil pH was measured using a suspension of air-dried soil and distilled water (in a ratio of 1:5) [24]. The soil organic matter (SOM) content was determined by the $K_2Cr_2O_7$-$H_2SO_4$ oxidation method [25]. Total nitrogen (TN) content was measured on a 2300 Kjeltec Analyzer Unit (FOSS, Höganäs, Sweden). Total phosphorus (TP) and total potassium (TK) were extracted according to literature [26]. We extracted and assayed available phosphorus (AP) and available potassium (AK) according to literature [27]. AP and AK was extracted from 2.5 g of soil using 25 mL solution of a mixture of 0.03 mol/L $NH_4F$ and 0.025 mol/L HCL and 25 mL solution of 1 mol/L $NH_4Ac$ (pH = 7.0), respectively. TP, TK, AP and AK contents were determined by ICAP (Spectro Analytical Instruments, Spectro Arcos ICP, Kleve, Germany).

### 2.3. Standing Litter Analyses

Five litter sampling spots of 1 m × 1 m were established within each plot. The litter layer was divided into a top organic layer (TOL) that mainly consisted of undecomposed litter and a bottom organic layer (BOL) that mainly consisted of decomposed litter according to the classification method proposed by Zanella et al. [28]. The dry weight of litter was measured after being oven-dried at 80 °C to constant weight.

### 2.4. Soil Enzyme Activities Analyses

Activities of the soil catalase (CAT), urease (UE), sucrase (SC) and β-glucosidase (BG) were determined by spectrophotometry [29]. Briefly, 1 g of soil was added to 9 mL of phosphate buffer solution (PBS, 0.01 mol·L$^{-1}$, pH = 7.2). The mixed solution was shaken well and then centrifuged at 4 °C for 20 min (2000 rpm). Finally, collected the supernatant. 10 uL of the supernatant was distributed to wells in a 96 wells plate, and the reagents were added according to the kit instructions (Qingdao Jiekangkang Biotechnology Co., Ltd., Qingdao, China). Samples below 37 °C were incubated for 60 min. Fluorescence at 450 nm was determined on a SpectraMax Paradigm Multi-Mode detection platform (Molecular devices, San Jose, CA, USA).

### 2.5. Soil Microbial Communities Analyses

Soil microbial DNA was extracted from fresh soil samples using the Power Soil DNA Isolation Kit (MoBio Laboratories, Carlsbad, CA, USA), following the manufacturer's instructions. The bacterial V3-V4 barcode region of bacterial and fungal ITS1 barcode region were amplified using forward primers (338F: 5′-GTACTCCTACGGG AGGCAGCA-3′, ITS1F: 5′-GGAAGTAAAAGTCGTAACAAGG-3′) and reverse primers (806R: 5′-GTGGACTACHVGGGTWTCTAAT-3′, ITS2: 5′-ATCCTCCGCTTATTGATATGC-3′), respectively. The polymerase chain reaction (PCR) parameters are as follows: denaturation at 95 °C for 2 min, 30 cycles of 95 °C for 30 s, 55 °C for 30 s and 72 °C for 30 s, and final extension at 72 °C for 10 min. The PCR products were purified by using a QIAquick Gel Extraction Kit (QIAGEN, Dusseldorf, Germany), and sequenced on Illumina MiSeq PE300 platform (Illumina, Inc., San Diego, CA, USA). Raw sequences were processed by Allwegene Tech, Ltd. (Beijing, China) using QIIME and MOTHUR. Briefly, the erroneous and chimeric sequences and pair-end reads with low-quality sequences score below 20 were removed. After trimming, sequences were clustered into operational taxonomic units (OTUs) at 97% similarity level using Uclust [30].

### 2.6. Statistical Analyses

An analysis of variance (ANOVA) was conducted using SPSS (IBM Corp., New York, NY, USA) and was used to evaluate the differences in the soil physicochemical properties, enzyme activities and abundance, diversity and composition of microbial community between different stand ages. Spearman's correlation analyses were used to explore the relationships between soil enzyme activities and microbial community diversity of different stand ages. Linear mixed models (LMM) were used to test for variables affecting the variance in each enzyme and microbial diversity index in the soil using physicochemical data as predictors, location and layer (nested within location) as random factors, and enzyme activities and microbial diversity indices as response variables [31]. To ensure normality and homoscedasticity, we applied a Box–Cox transformation with $\lambda = 0.5$ for dependent variables. To quantify unbiased measurements of variance expressed by fixed and fixed + random factors, respectively [32], we used conditional and marginal determination coefficients ($R^2$c and $R^2$m) to express the multilevel model goodness of fit. The R packages 'lme4' and 'sjPlot' were used to conduct the analyses.

## 3. Results

### 3.1. Soil Physicochemical Properties

Differences in soil SBD, pH, SOM, TN, TP, TK, AN, AP, and AK contents were found among the seven stand ages and among three depths (Figure 2). We observed that SBD values were influenced by stand age and soil depth, with higher values in high stand age and deep depth soil than those from low stand age and shallow depth soil. Stand age affected soil pH and created an order of 49a > 32a > 25a > 18a > 12a > 3a > 6a. The pH values were affected by depth, higher for subsurface than surface. As stand age increased, the contents of SOM, TN, TP, TK, AN, AP and AK gradually decreased first, then gradually increased, and were the highest in 49a and lowest in 12a or 18a. As soil depth increased, the contents of TN, TP, TK, AN, AP and AK gradually decreased. In particular, soil nutrients including SOM, TN, AN, AP, and AK contents in D1 were significantly higher than those in D2 and D3. The ratio of carbon to nitrogen (C:N) values first increased gradually from 6a to 18a, and then decreased gradually from 18a to 49a. The C:N values at D1 and D2 in 3a were higher than 6a. As the soil depth increased, the C:N value showed an uptrend. Interestingly, the C:N value was always similar at D2 and D3 depths in the seven stand age plantations.

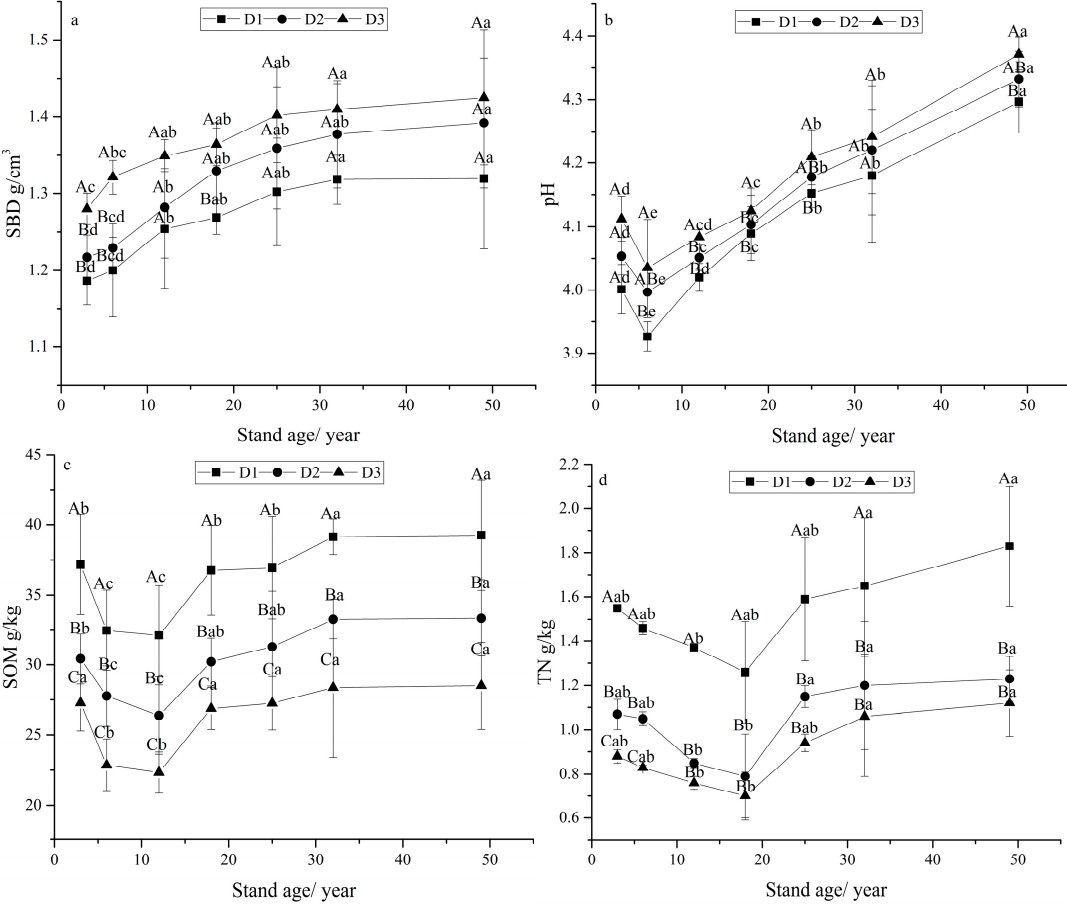

**Figure 2.** *Cont.*

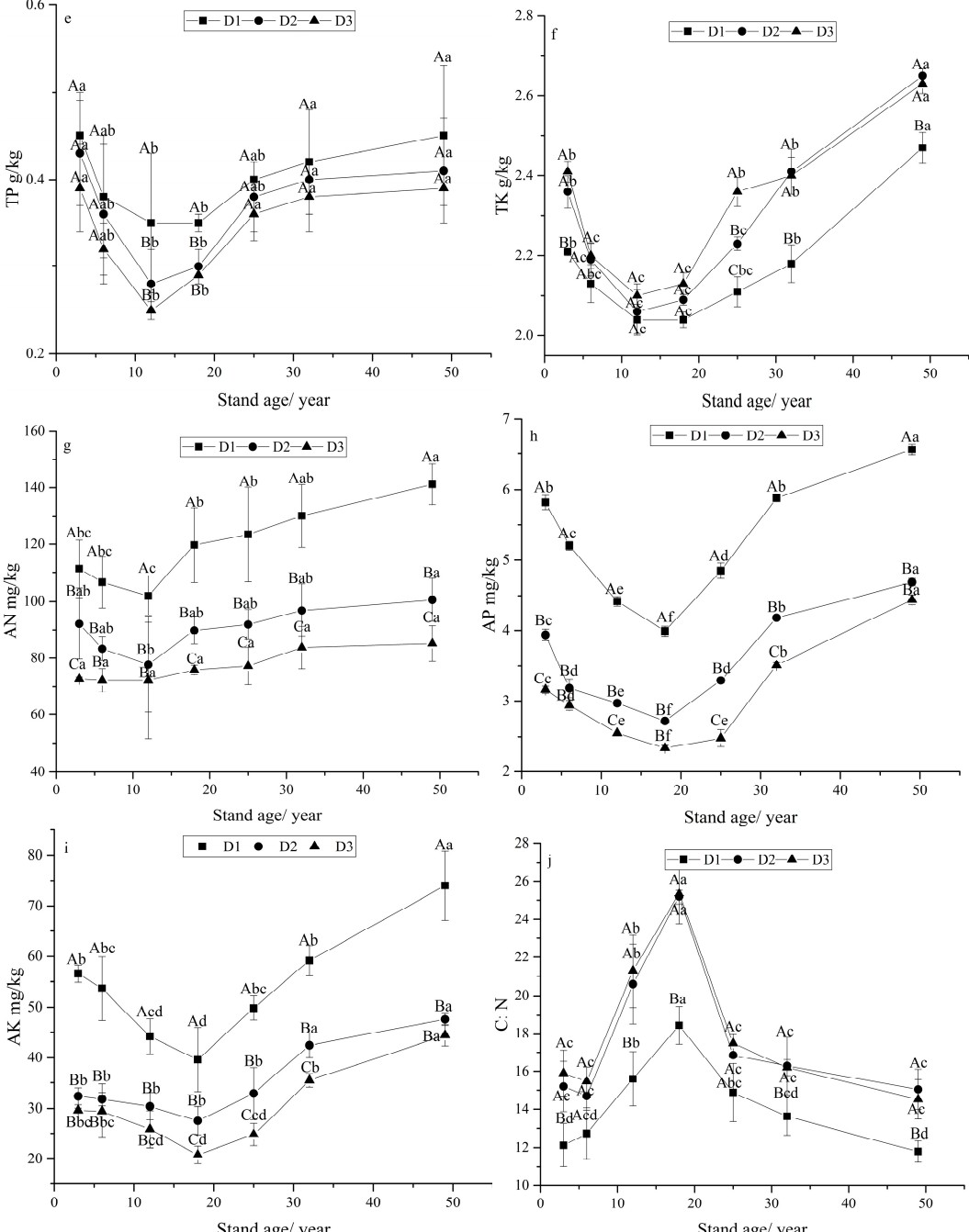

**Figure 2.** Soil physicochemical characteristics in the different age soils. (**a**) soil bulk density (SBD) values in different stand ages and different soil depths, (**b**) pH values in different stand ages and different soil depths, (**c**) soil organic matter (SOM) contents in different stand ages and different soil depths, (**d**) total nitrogen (TN) contents in different stand ages and different soil depths, (**e**) total phosphorus (TP) contents in different stand ages and different soil depths, (**f**) total potassium (TK) contents in different stand ages and different soil depths, (**g**) alkali hydrolysis nitrogen (AN) contents in different stand ages and different soil depths, (**h**) available phosphorus (AP) contents in different stand ages and different soil depths, (**i**) available potassium (AK) contents in different stand ages and different soil depths, (**j**) the ratio of carbon to nitrogen (C: N) values in different stand ages and different soil depths. Different lowercase letters and different uppercase letters represent significant differences between the same soil depth (D1: 0–10, D2: 10–20 or D3: 20–30 cm) of different stand ages and between different soil depths (D1: 0–10, D2: 10–20 and D3: 20–30 cm) of the same stand age at $p < 0.05$, respectively.

### 3.2. Standing Litter Stocks

In the seven stand-age plantations, the litter stocks at the TOL layer followed an order from 32a > 49a > 25a > 18a > 3a > 12a > 6a, while the litter stocks at the BOL layer and total litter stocks created an order of 49a > 32a > 25a > 18a > 3a > 12a > 6a (Table 2). The litter stock in 6a was significantly lower than in the other six stand-age plantations ($p < 0.05$, Table 2). The litter stock was similar in 3a and 12a (Table 2). The litter stocks in 32a and 49a were significantly higher than those in the other five stand-age plantations ($p < 0.05$, Table 2). The differences in the litter stocks among 12a, 18a and 25a were significant ($p < 0.05$, Table 2). The ratio of litter stock at the BOL layer to total litter stock was lowest in 3a and gradually increased from 12a to 49a, while the ratio of litter stock at the TOL layer to total litter stock was the opposite (Table 2).

**Table 2.** Standing litter stocks in the different age soils.

| Stand Age (a) | Litter Stocks/ (kg·hm$^{-2}$) | | | Ratio of Litter Stock at Different Layer to Total Litter Stock/% | |
|---|---|---|---|---|---|
| | TOL | BOL | Total | TOL | BOL |
| 3a | 5864 ± 202 d | 3281 ± 116 e | 9146 ± 225 d | 64.11 | 35.89 |
| 6a | 3155 ± 168 e | 1982 ± 118 f | 5137 ± 122 e | 61.39 | 38.61 |
| 12a | 5261 ± 138 d | 3083 ± 88 e | 8343 ± 162 d | 63.05 | 36.95 |
| 18a | 8590 ± 187 c | 5674 ± 162 d | 14264 ± 250 c | 60.22 | 39.78 |
| 25a | 11073 ± 822 b | 9066 ± 789 c | 20139 ± 808 b | 54.98 | 45.02 |
| 32a | 13201 ± 598 a | 11944 ± 386 b | 25145 ± 774 a | 52.48 | 47.52 |
| 49a | 12495 ± 1367 a | 13108 ± 817 a | 25604 ± 2000 a | 48.71 | 51.29 |

Note: Abbreviations: TOL: top organic layer, BOL: bottom organic layer. Significant differences between the seven age soils were determined using one-way analysis of variance (ANOVA) at $p < 0.05$. The data are shown as the means ± standard deviation (SD) ($n = 9$). Different lowercase letters represent significant differences between different stand ages at $p < 0.05$.

### 3.3. Soil Enzyme Activities

In this study, we found that evaluated CAT, UE, SC and BG activities all were influenced significantly by stand age and soil depth (Figure 3). Overall, soil depth significantly reduced the four enzyme activities. Stand age significantly increased soil CAT activity, whereas stand age first had negative effects on UE, SC and BG activities from 6a to 18a, and then had positive effects on SC and BG activities and UE activities in D1 and D2 from 18a to 49a. The SC and BG activities in 3a were lower than those in 6a, while the UE activity was opposite. The UE activity was lowest in the subsurface soil samples collected from 25a at D3.

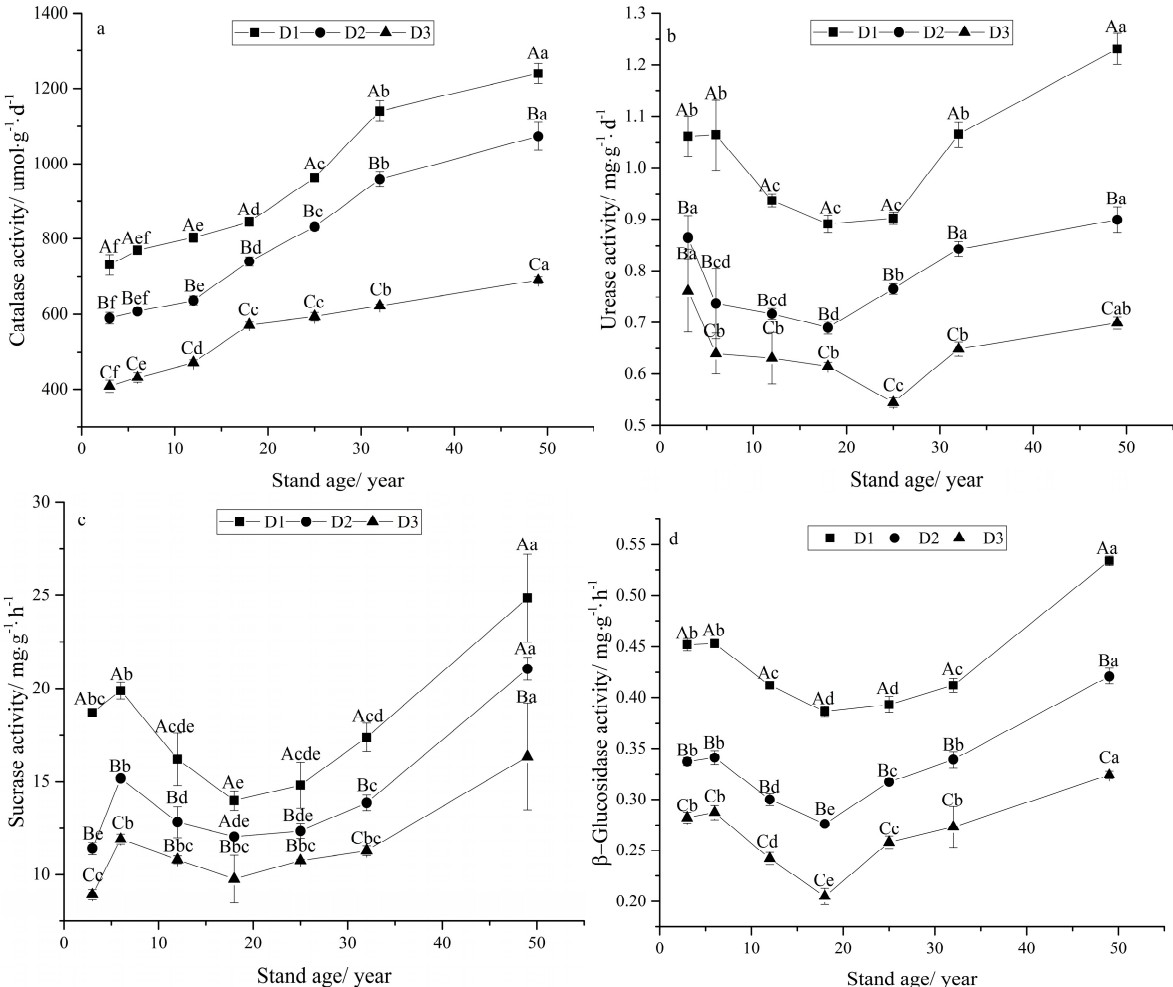

**Figure 3.** The soil enzyme activities of plantations of different ages. (**a**) catalase activities in different stand ages and different soil depths, (**b**) urease activities different stand ages and different soil depths, (**c**) sucrase activities different stand ages and different soil depths, (**d**) β-glucosidase activities in different stand ages and different soil depths. Different lowercase letters and different uppercase letters represent significant differences between the same soil depth of different stand ages and between different soil depths of the same stand age at $p < 0.05$, respectively.

### 3.4. Microbial Communities

Microbial community profiling showed 25, 25 and 27 bacterial phyla, 12, 12 and12 fungal phyla, 287, 249 and 248 bacterial genera, and 431, 321 and 328 fungal genera in D1, D2 and D3 of seven stand ages, respectively. The Shannon index indicated that the microbial community diversity in three depths was significantly different among 18a and 49a and among 25a and 49a ($p < 0.05$), and the richness index for Chao1 was similar (Table 3). The richness and diversity of bacterial and fungal communities first decreased gradually from 6a to 18a or 25a, and then increased from 18a or 25a to 49a. The differences in the richness and diversity of bacterial and fungal communities were not significant between 3a and 6a and between 18a and 25a, although the stand age can drive the changes in these indices. The richness and diversity of microbial communities were influenced significantly by depth ($p < 0.05$). In particular, the values of Shannon and Chao1 indices in D1 were significantly higher than D2, except for bacterial Chao1 index values in D1 and D2 of 6a, 12a, 18a, 25a and 49a, whereas the community richness exhibited no significant difference between D2 and D3 of seven stand ages except for fungal Chao1 in 25a.

**Table 3.** Shifts in richness and diversity of microbial communities in different stand ages.

| Stand Age (a) | Bacteria | | | | | | Fungi | | | | | |
| --- | --- | --- | --- | --- | --- | --- | --- | --- | --- | --- | --- | --- |
| | Shannon | | | Chao1 | | | Shannon | | | Chao1 | | |
| | D1 | D2 | D3 | D1 | D2 | D3 | D1 | D2 | D3 | D1 | D2 | D3 |
| 3a | 9.99 ± 0.12 Aab | 9.49 ± 0.19 Bab | 9.24 ± 0.09 Babc | 3864 ± 174 Aab | 3599 ± 207 Bab | 3412 ± 159 Bab | 6.47 ± 0.16 Abc | 5.93 ± 0.03 Bb | 5.85 ± 0.24 Bb | 820 ± 41 Abc | 534 ± 13 Bbc | 517 ± 59 Bb |
| 6a | 10.03 ± 0.11 Aab | 9.53 ± 0.02 Bab | 9.30 ± 0.19 Bab | 4194 ± 104 Aa | 3800 ± 428 ABab | 3475 ± 283 Bab | 7.14 ± 0.24 Aab | 6.00 ± 0.38 Bb | 5.95 ± 0.49 Bb | 856 ± 23 Aabc | 566 ± 12 Bab | 536 ± 91 Bab |
| 12a | 9.97 ± 0.09 Aab | 9.42 ± 0.15 Bab | 9.02 ± 0.31 Bbc | 3997 ± 192 Aab | 3655 ± 98 ABab | 3316 ± 173 Bab | 6.62 ± 0.02 Aabc | 5.89 ± 0.06 Bb | 5.63 ± 0.18 Bbc | 854 ± 38 Abc | 492 ± 46 Bcd | 447 ± 27 Bc |
| 18a | 9.62 ± 0.12 Ab | 9.05 ± 0.11 Bc | 8.99 ± 0.07 Bbc | 3496 ± 336 Aab | 3246 ± 47 Ac | 3155 ± 213 Ab | 6.53 ± 0.10 Aabc | 5.83 ± 0.16 Bb | 5.26 ± 0.14 Bc | 765 ± 10 Ac | 431 ± 27 Bd | 350 ± 27 Bd |
| 25a | 9.70 ± 0.45 Ab | 9.25 ± 0.05 Bbc | 8.94 ± 0.03 Cc | 3219 ± 174 Ab | 3440 ± 19 Abc | 2911 ± 440 Ab | 6.06 ± 0.79 Ac | 5.81 ± 0.40 ABc | 5.32 ± 0.12 Bc | 785 ± 104 Abc | 559 ± 20 Bb | 440 ± 32 Cc |
| 32a | 10.08 ± 0.05 Aab | 9.32 ± 0.08 Bab | 9.29 ± 0.01 Babc | 4202 ± 108 Aa | 3490 ± 264 Bbc | 3472 ± 134 Bab | 7.14 ± 0.23 Aab | 6.33 ± 0.21 Bab | 5.93 ± 0.32 Bb | 887 ± 83 Aab | 587 ± 18 Bab | 548 ± 34 Bab |
| 49a | 10.16 ± 0.15 Aa | 9.55 ± 0.07 Ba | 9.47 ± 0.11 Ba | 4306 ± 450 Aa | 4024 ± 235 Aa | 3507 ± 187 Ba | 7.20 ± 0.01 Aa | 6.70 ± 0.09 Ba | 6.63 ± 0.24 Ba | 966 ± 87 Aa | 609 ± 49 Ba | 594 ± 63 Ba |

Note: D1: 0–10 cm, D2: 10–20 cm, D3: 20–30 cm. Significant differences between the seven age soils were determined using one-way ANOVA at $p < 0.05$. The data are shown as the means ± SD ($n = 9$). Different lowercase letters and different uppercase letters represent significant differences between the same soil depth of different stand ages and between different soil depths of the same stand age at $p < 0.05$, respectively.

**Table 4.** Results from the linear mixed model (LMM) analysis of enzyme activities and microbial diversities. Response variables are reported in columns, while predictors (both fixed and random parts) are reported in rows. Location as random factor, layer as nested factor within location. Statistical significance is reported as * ($p < 0.050$), ** ($p < 0.010$), *** ($p < 0.001$).

| | Catalase | Urease | Sucrase | β-glucosidase | Bacteria | | Fungi | |
| --- | --- | --- | --- | --- | --- | --- | --- | --- |
| | | | | | Shannon | Chao1 | Shannon | Chao1 |
| | Estimate (CI) | Estimate (CI) | Estimate (CI) | Estimate (CI) | Estimate (CI) | Estimate (CI) | Estimate (CI) | Estimate (CI) |
| **Fixed Parts** | | | | | | | | |
| Intercept | 0.42 (−0.39–1.23) *** | 0.71 (0.37–1.05) *** | 0.76 (−1.22–0.3) *** | 0.14 (0.01–0.27) *** | 1.35 (1.09–1.61) *** | 2.09 (2.05–2.13) *** | 2.96 (1.1–4.82) *** | 1.74 (1.65–1.83) *** |
| SBD | 0.00 (−0.02–0.02) *** | 0.07 (−0.03–0.17) | 0.33 (−0.86–2.18) | 0.05 (0.01–0.09) ** | 0.08 (−0.29–0.45) | 0.64 (0.12–1.16) | 0.78 (0.03–1.53) * | 0.37 (−0.34–1.08) |
| pH | 0.99 (0.98–1.00) | −0.11 (−0.21–0.01) | 0.31 (−0.09–0.71) | −0.01 (−0.05–0.03) | 0.07 (−0.28–0.42) | −0.48 (−1.11–0.15) | −0.45 (−1.02–0.12) * | −0.35 (−1.07–0.37) ** |
| SOM | 0.48 (−0.23–1.19) | 0.49 (−0.21–1.19) | −0.02 (−0.05–0.01) | 0.01 (−0.01–0.03) | 0.00 (−0.02–0.02) | −0.81 (−1.23–0.39) | −0.99 (−1.00–0.98) * | −0.76 (−1.21–0.31) |
| TN | 0.99 (0.98–1.00) | 0.01 (−0.03–0.05) * | 0.43 (−0.93–1.79) * | 0.00 (−0.02–0.02) | 0.22 (0.06–0.38) ** | 0.19 (−0.66–1.04) ** | −0.18 (−0.47–0.11) *** | −0.67 (−0.77–0.57) * |
| TP | −0.58 (−1.14–0.02) | −0.14 (−0.30–0.02) | −0.59 (−0.86–0.32) | −0.03 (−0.09–0.03) | −0.47 (−0.08–0.86) | −0.78 (−1.22–0.34) | −0.12 (−0.28–0.04) | −0.08 (−1.02–0.86) |
| TK | 0.43 (−0.34–1.20) | 0.68 (0.27–1.09) | 0.63 (−0.09–1.35) | 0.00 (−0.01–0.01) | 0.02 (−0.07–0.11) | −0.86 (0–1.25–0.47) | −0.4 (−0.58–0.22) | −0.67 (−1.17–0.17) |
| AN | 0.72 (0.36–1.08) | 0.06 (−0.04–0.16) | 0.00 (−0.02–0.02) | 0.02 (0.00–0.04) | 0.00 (−0.01–0.01) | −0.45 (−0.60–0.30) | 0.75 (0.71–0.79) ** | 0.74 (0.27–1.21) |
| AP | 0.01 (−0.55–0.57) * | 0.00 (−0.09–0.09) *** | 0.66 (0.76–0.56) *** | 0.04 (0.03–0.05) *** | 0.39 (0.29–0.49) *** | 0.62 (0.61–0.63) *** | 0.27 (0.1–0.44) *** | 0.11 (0.07–0.15) *** |
| AK | −0.13 (−0.69–0.43) | 0.88 (0.72–1.04) | 0.01 (−0.05–0.07) | 0.08 (0.07–0.09) | −0.01 (−0.02–0.00) | −0.67 (−1.17–0.17) | 0.00 (−0.01–0.01) | 0.82 (0.40–1.24) |
| C:N | −0.77 (−1.07–0.47) | 0.61 (0.09–1.13) | −0.02 (−0.07–0.03) | 0.02 (−0.01–0.05) | 0.00 (−0.01–0.01) | −0.77 (−1.21–0.33) | −0.01 (−0.02–0.00) | −0.3 (−0.33–0.27) * |
| **Random Parts** | | | | | | | | |
| $\sigma^2$ | 0.016 | 0.003 | 0.085 | 0.001 | 0.335 | 0.050 | 0.025 | 0.005 |
| $\tau_{00,\ Location}$ | 0.031 | 0.102 | 0.152 | 0.264 | 0.241 | 0.252 | 0.312 | 0.335 |
| $\tau_{00,\ Layer:Location}$ | 0.085 | 0.142 | 0.114 | 0.251 | 0.284 | 0.302 | 0.216 | 0.284 |
| $ICC_{Location}$ | 0.235 | 0.413 | 0.433 | 0.512 | 0.280 | 0417 | 0.564 | 0.537 |
| $ICC_{Layer:Location}$ | 0.644 | 0.575 | 0.325 | 0.486 | 0.330 | 0.500 | 0.391 | 0.455 |
| $R^2c, R^2m$ | 0.085, 0.284 | 0.101, 0.425 | 0.176, 0.378 | 0.129, 0.439 | 0.185, 0.537 | 0.127, 0.485 | 0.238, 0.685 | 0.204, 0.641 |

Abbreviations: SBD: Soil bulk density, SOM: soil organic matter, TN: total nitrogen, TP: total phosphorus, TK: total potassium, AN: alkali hydrolysis nitrogen, AP: available phosphorus, AK: available potassium, C:N: the ratio of carbon to nitrogen, σ2: variance, $\tau_{00}$: covariance, ICC: intraclass correlation coefficient, $R^2c$: conditional determination coefficients, $R^2m$: marginal determination coefficients.

The bacterial profiles of different stand ages were dominated by Acidobacteria, Proteobacteria, Actinobacteria, Chloroflexi, and Firmicutes (Figure 4). These five dominant bacterial phyla accounted for more than 92% of the total bacterial communities in all samples. However, their relative abundance varied with the increase of stand age. When stand age first increased from 3a to 6a, 12a, 18a and 25a, the abundance of Acidobacteria at D1 increased from 37.72% to 43.42%, 45.40%, 51.55% and 53.36%, respectively, and then decreased from 25a to 32a and 49a; the abundance of Acidobacteria at D1 decreased from 53.36% to 44.91% and 33.46%, respectively (Figure 4a). However, at the depth of D2, the relative abundance of Acidobacteria gradually increased from 3a to 12a, and then gradually decreased from 12a to 49a (Figure 4b), while, at the depth of D3, it gradually increased from 3a to 18a, and then gradually decreased from 18a to 32a (Figure 4c). The relative abundance of Acidobacteria at D1 in 18a and 25a was significantly higher than those in 3a and 49a ($p < 0.05$, Table S1). At the depth of D2, no significant differences in relative abundance of all Acidobacteria between seven stand-age soils were observed ($p > 0.05$, Table S1). At the depth of D3, the relative abundance of Acidobacteria in 6a, 12a, and 18a was significantly higher than that in 3a, 32a, and 49a ($p < 0.05$, Table S1). At the three soil depths, relative abundance of Proteobacteria all tended to first decrease when the stand age increased from 3a to 18a and then increase when stand age increased from 18a to 49a (Figure 4). Despite this, the proportion of Proteobacteria in low stand age (3a and 6a) was higher than in high stand age (25a, 32a and 49a) and significantly higher than that in 18a and 25a ($p < 0.05$, Table S1). Although significant differences in relative abundance of Actinobacteria, Chloroflexi, and Firmicutes were observed among stand-age plots, these changes have no obvious regularity (Figure 4). Overall, the relative abundance of Acidobacteria, Chloroflexi, and Firmicutes gradually increased with the increase of soil depth except for the proportion of Acidobacteria in 18a, 25a and 32a, and the proportion of Firmicutes in 18a and 25a (Table S1). While the relative abundance of Proteobacteria and Actinobacteria gradually decreased with the increase of soil depth except for the proportion of Actinobacteria in 18a and 25a (Table S1).

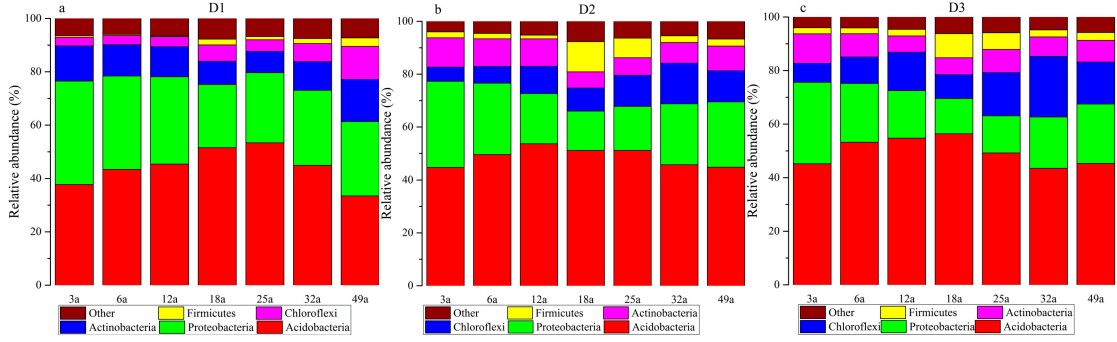

**Figure 4.** The compositions of bacterial at the phylum level in different stand age samples. (**a**) Bacterial community at the phylum level in D1, (**b**) bacterial community at the phylum level in D2, (**c**) bacterial community at the phylum level in D3.

Ascomycota, Basidiomycota and Zygomycota were the diminant fungal phyla in all samples, accounting for more than 95% of the total fungal community except for 87.02% of the total fungal community at D1 in 25a (Figure 5). The relative abundance of Ascomycota at D1 first decreased gradually from 6a to 25a, and then increased from 25a to 49a, while at D2 and D3 it first decreased gradually from 6a to 18a, and then increased gradually from 18a to 49a (Figure 5). Interestingly, the relative abundance of Ascomycota at three depths in 3a was lower than that in 6a (Figure 5). The relative abundance of Ascomycota in 49a was markedly higher than that in 18a ($p < 0.05$, Table S1). Although the changes in the relative abundance of Basidiomycota and Zygomycota had no obvious regularity with the increase of stand age, the difference in their relative abundance was significant between different stand ages, and the changes of these fungal phyla with soil depth were similar to this result (Table S1).

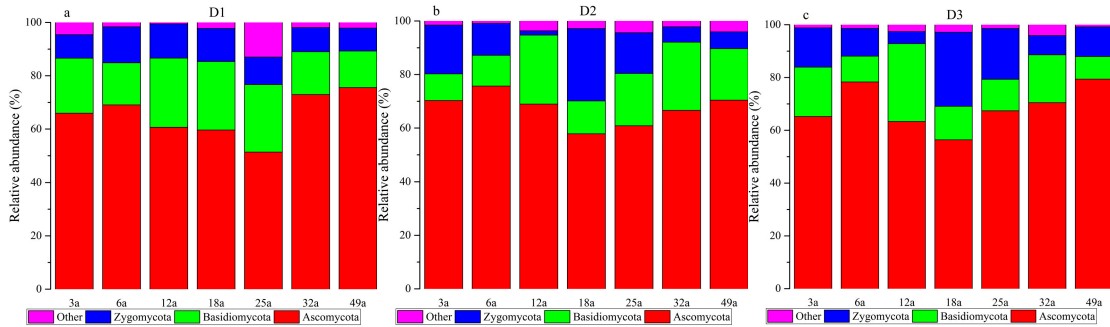

**Figure 5.** The compositions of fungal community at the phylum level in different stand age samples. (**a**) Fungal community at the phylum level in D1, (**b**) fungal community at the phylum level in D2, (**c**) fungal community at the phylum level in D3.

### 3.5. Links Between Soil Physicochemical Properties, Enzyme Activities and Microbial Diversity Indices

The LMM results showed that the conditional determination coefficients ($R^2$c) for enzyme activities and microbial diversity indices all were lower than the marginal determination coefficients ($R^2$m), showing that location and layer had a strong effect in the model owing to the fact that the greater the difference between these two values, the greater the contribution to the overall variance given by the random effects (Table 4). Within the fixed parts, only AP was a significant predictor for all response variables ($p < 0.05$). TN proved to be significant for all response variables except for catalase and β-glucosidase, and SBD proved to be significant for Catalase, β-glucosidase and fungal Shannon index ($p < 0.05$). Our results showed that pH was a significant predictor for fungal Shannon and Chao1 indices, SOM and AN were significant predictors for the fungal Shannon index, and C:N was a significant predictor for the fungal Chao1 index ($p < 0.05$). Positive correlations between soil enzyme activities and microbial diversity indices were significant ($p < 0.01$, Table 5).

**Table 5.** Spearman's correlation between soil enzyme activities and microbial diversity indices.

| Enzymes | Bacteria | | Fungi | |
| --- | --- | --- | --- | --- |
| | Shannon | Chao1 | Shannon | Chao1 |
| CAT | 0.545 ** | 0.404 ** | 0.642 ** | 0.676 ** |
| UE | 0.793 ** | 0.601 ** | 0.732 ** | 0.833 ** |
| SC | 0.727 ** | 0.570 ** | 0.756 ** | 0.721 ** |
| BG | 0.833 ** | 0.663 ** | 0.798 ** | 0.851 ** |

Note: Abbreviations: CAT: catalase, UE: urease, SC: sucrase, BG: β-glucosidase. ** Shows significant differences at $p < 0.01$.

## 4. Discussion

### 4.1. Shift in Soil Pyhsicochemical Properties and Standing Litter Stocks

In the present study, we found significant differences in soil properties among seven stand ages of Chinese fir plantations and among three soil depths under seven stand ages of Chinese fir plantations. The result showed that the stand age had a significantly positive effect on SBD (Figure 2a), which is not in line with the results of Zhu and Meng [33] who reported SBD values in 26-35a were lower than in 21–25a. As forest aged, the increase in rainfall and artificial tending time may be the main reasons for the gradual increase in SBD [34,35]. The contents of TN, TP, TK, AN, AP and AK in 12a and 18a all were lower than those in the other five stand ages, which is inconsistent with the research results of Li et al. [36]. As we all know, soil nutrient content is related to litter stock. In this study, although the litter stock in 6a was lower than 12a, the nutrient contents in 3a and 6a were higher than 12a, mainly because 10–20-year-old Chinese fir was in the rapid growth stage and assimilated a large amount of nutrients from the soil. However, the density of Chinese fir plantation decreased, the growth rate

slowed down, the nutrient uptake decreased, and the litter stocks increased from 18a to 49a, which may have led to the increase of soil nutrient content. The C:N ratio is an important indicator of soil C and N mineralization. The lower the C:N, the more favorable the release of N, and low rates of N release occur when C:N ratio exceeds 15 [37]. In this study, C:N ratio in 18a was highest, indicating that C and N cycle efficiency and soil microbial metabolic activity were lower than the other stand ages. Although soil pH is related to organic matter content and microbial metabolism [38,39], it is necessary to further study why the pH gradually rises from 6a to 49a. Vertical differences in soil nutrient contents were mainly due to the decrease in organic matter content and microbial diversity.

*4.2. Shift in Soil Enzyme Activities and Composition and Diversity of Microbial Communities*

Our results supported the first and partially supported the third hypothesis, soil enzyme activities changed significantly across stand age, but stand age had not a positive or a negative effect on UE, SC and BG activities (Figure 3). Soil enzyme activity is closely related to microorganisms because soil enzymes are mainly secreted by microorganisms and plant roots [40,41], which is also supported by our results of the correlations between enzyme activities and microbial diversity indices. In our study, overall, the activities of UE, SC and BG in 12a, 18a and 25a were lower than 6a, 32a and 49a, which is in line with the trend of microbial diversity indices (Figure 3, Table 3), indicating that these enzyme activities in Chinese fir plantations may be influenced by microorganisms. Although we find a correlation between microbial community diversity and enzyme activity, we note that this study does not reveal if this connection is simply a co-correlation to another soil factor, or a mechanism that may explain patters in enzyme activity. Further research is needed to address this question. Contrary to the third hypothesis, we found that the stand age of Chinese fir plantations had a positive influence on CAT activity (Figure 3a). However, Shu [18] reported that the CAT activity at 0–10 cm depth created an order of 8a> 11a> 15a, while at 10–20 cm and 20–30 cm depths it created an order of 8a > 15a > 11a. Gu et al. [42] reported that the decrease in stand density can stimulate CAT activity, which may partly explain the increase in CAT activity. Vertical changes in soil enzyme activities were found in many studies [43,44].

In support of the second and the third hypothesis, the results showed that the Shannon and Chao1 of bacterial and fungal communities in 3a were all lower than those in 6a, mainly because controlled burning before afforestation destroyed the composition and structure of the microbial community. We found that the richness and diversity of the microbial community in 18a and 25a were lower than those in the other five stand ages, which is different from the results of other studies. Wei et al. [45] reported that the richness and diversity of the soil bacterial community at 0–10 cm depth of Chinese fir plantation followed an order from 10a > 20a > 38a. Liu et al. [20] found that the richness and diversity of the microbial community at 0–20 cm depth of Chinese fir plantation created an order of 37a > 24a > 21a > 17a > 11a > 3a using Denaturing Gradient Gel Electrophoresis (DGGE). The decrease in the richness and diversity of soil microbial communities with the increase of stand age from 6a to 18a or 25a was related to the decrease in soil nutrient content and diversity of undergrowth vegetation. As stand age increases from 25a to 49a, there is an increase in the composition and diversity of undergrowth vegetation, and the amount of litter fall and nutrient contents may have promoted the increase of richness and diversity of microbial community.

The microbial compositions of Chinese fir plantations of different ages were different. The bacterial dominance of Acidobacteria, Proteobacteria, Actinobacteria, Chloroflexi, and Firmicutes and the fungal dominance of Ascomycota, Basidiomycota and Zygomycota in the present study have been observed in various soils types [45–49], indicating that these microbial phyla play important roles in soil ecosystems. Changes in microbial community composition indicate changes in soil ecological environment and function [50,51]. Acidobacteria can accelerate the conversion of nitrates and nitrites [52], which is supported by our findings because the increase in the abundance of Acidobacteria is accompanied by a decrease in TN and AN contents. The relative abundance of Proteobacteria is closely related to soil carbon [48]. The lower Proteobacteria abundance in 18a and 25a and the decrease of the relative abundance of Proteobacteria with soil depth may be owing to its lifestyle, because the relative

abundance of Proteobacteria is higher in soil with higher carbon availability [53]. Given the strong correlation to organic substrates [54], the decrease in the relative abundance of Ascomycota from 6a to 18a or 25a and the increase from 25a to 49a may be related to litter fall stocks and SOM content according to the correlations revealed in this study.

*4.3. Effects of Soil Physicochemical Properties on Enzyme Activities and Microbial Diversity*

Previous studies reported that soil chemical properties such C, N, P and K had significant effects on microbial community [55,56] Many studies suggested that soil enzyme activities and microorganisms are influenced by the soil pH [53,57]. In this study, however, TN and AP contents were primary influencing factors and more important than soil pH in terms of soil enzyme activities and microorganisms responses to stand-age change drivers. The LMM results showed that enzyme activities and microbial community diversities were also affected by random factors, indicating that the use of the LMM is more conducive to determining predictors of enzyme activities and microbial communities. We found that the correlation of the richness and diversity of soil microbial community with soil enzyme activities was significant (Table 5). Therefore, changes in soil microbial community composition caused by shifts in stand age may indirectly affect soil enzyme activities. However, further research is needed to understand whether this correlation is related to some other variable or is a direct relationship.

## 5. Conclusions

Our study demonstrated that the stand age of Chinese fir plantations can drive changes in soil physicochemical properties, enzyme activities and the richness and diversity of the microbial community at 0–30 cm depth. Low stand age (3a and 6a) and high stand age (32a and 49a) showed high-nutrient content, the activities of UE, SC and BG, and the richness and diversity of microbial community compared to middle stand age (12a, 18a and 25a). Given that the stand age-driven changes in soil biotic and abiotic properties observed in this study are common to many plantations, our findings provide important insights into the impact of stand age on soil microbial communities and enzyme activities. In addition, we observed that changes in the richness and diversity of soil microbial communities across stand age correlated with enzyme activities. Given the connection between soil enzyme activity and microbial communities, our research suggests the need to further understand how soil microbial diversity may drive enzyme activity within forest management.

**Supplementary Materials:** The following are available online at http://www.mdpi.com/1999-4907/10/1/11/s1, Table S1: Shifts in relative abundance of the dominated bacterial and fungal communities at phylum level at different depths in different stand ages.

**Author Contributions:** Conceptualization, C.W. and R.J.; Methodology, C.W. and R.J.; Software, C.W. and L.X.; Validation, C.W., L.X., Y.D., L.H., Y.W., J.C. and R.J.; Formal Analysis, C.W. and L.X.; Investigation, C.W., Y.D., and R.J.; Resources, C.W. and R.J.; Data Curation, C.W.; Writing—Original Draft Preparation, C.W.; Writing—Review and Editing, C.W., L.X., and R.J.; Visualization, C.W.; Supervision, R.J.; Project Administration, C.W.; Funding Acquisition, R.J.

**Funding:** This research was funded by State Key Research Development Program of China (Grant number 2016YFD0600300). This study is a part of the project entitled "Study on efficient cultivation technology of Chinese fir plantation".

**Acknowledgments:** This work is supported by the State Key Research and Development Program of China (2016YFD0600300).

**Conflicts of Interest:** The authors declare no conflict of interest.

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
