# Peer review of "Contrasting Effects of Chinese Fir Plantations of Different Stand Ages on Soil Enzyme Activities and Microbial Communities"

_forests, doi:10.3390/f10010011_

Round 1
Reviewer 1 Report
This study is straight-forward, well scoped study of the differences in soil properties (edaphic factors, microbial communities, and enzyme activity) across different stand ages of a managed forest. The replication is nicely set up to reveal patterns to this end. I believe that this study is of interest to the readership of forests, but need some revisions below.
1) In general the manuscript needs to further qualify some of its findings. For example, you should talk about changes across stand age, but qualify the statement that stand age drove these. Try to soften the language around this- suggestions below.
2) The connection between enzymes and microbial community diversity needs to be better qualified. In this study you find interesting correlations between the two but this does not actually show mechanism. So you need to throughout the manuscript simply indicate this correlation. I have added comments about places where caveats should be included.
3) The introduction needs to be expanded. There needs to be better framing within current ecological understandings of plant-soil microbe- enzyme interactions, and better connection to existing understanding of how increasing forest/stand age may impact soil fertility/biogeochemistry. See below for recommendation:
Before line 43: 1) Start with general discussion of plant-microbe interaction and current knowledge, one type of suggest is something like:
Plant-microbe interactions are central to soil fertility and ecosystem function. In both natural and managed ecosystems research has demonstrated the strong selective effects of plants on soil microbial communities and how resultant soil microbial communities may directly impact plant communities and soil nutrient cycling (Van Der Heijden et al. 2008, Bueno de Mesquita 2017, Yuan et al. 2017)
+ Van Der Heijden, Marcel GA, Richard D. Bardgett, and Nico M. Van Straalen. "The unseen majority: soil microbes as drivers of plant diversity and productivity in terrestrial ecosystems." Ecology letters 11.3 (2008): 296-310.
+ de Mesquita, Clifton P. Bueno, et al. "Plant colonization of moss-dominated soils in the alpine: Microbial and biogeochemical implications." Soil Biology and Biochemistry111 (2017): 135-142.
+ Yuan, Xia, et al. "Plant community and soil chemistry responses to long‐term nitrogen inputs drive changes in alpine bacterial communities." Ecology 97.6 (2016): 1543-1554.
Before line 51: You can connect these understandings more generally to your research, again cite relevant research in the field. Describe how increasing forest age may implicate microbial communities and the importance of these connections to ecosystem function. For example, add something like the following:
In general, studies of ecosystem succession have been central in demonstrating that forest ecosystems of increasing age can impact microbial community structure in variable ways over time. For example, there can be increasing selective effects of plant communities on microbial community structure and related function (Knelman et al. 2017, 2018 ). As microbial communities mediate biogeochemical cycles associated with nutrient and carbon cycling, in managed forest ecosystems the influence of increasing stand age on microbial communities may have strong effects on ecosystem functioning and soil fertility via soil enzyme activity (Holden et al. 2013, Smith et al 2015).
+Knelman, Joseph E., et al. "Interspecific Plant Interactions Reflected in Soil Bacterial Community Structure and Nitrogen Cycling in Primary Succession." Frontiers in microbiology 9 (2018): 128.
+Knelman, Joseph E., et al. "Rapid shifts in soil nutrients and decomposition enzyme activity in early succession following forest fire." Forests 8.9 (2017): 347.
+ Holden, Sandra R., Abraham Gutierrez, and Kathleen K. Treseder. "Changes in soil fungal communities, extracellular enzyme activities, and litter decomposition across a fire chronosequence in Alaskan boreal forests." Ecosystems 16.1 (2013): 34-46.
+Smith, A. Peyton, Erika Marín‐Spiotta, and Teri Balser. "Successional and seasonal variations in soil and litter microbial community structure and function during tropical postagricultural forest regeneration: a multiyear study." Global change biology 21.9 (2015): 3532-3547.
line 44: What is meant by 'energy flow' . Do you just mean soil nutrient and carbon cycling? Please revise.
line 65: agreed UPON
line 75/76: instead of "stand age will change" I would say "stand age will impact," because change more strongly suggests direct effects, while effects may be indirect.
Line 83/84: each plantation was given three plots WITHIN EACH BLOCK.
Line 89: soil samples WERE stored at...
Line 90: They were sieved again? In what way. Or if this is the same as the 2mm mentioned above please make sure the chronology of what you did is correctly communicated.
Line 91: How are these stands managed, if at all. Is there any thinning. Is there any fertilization or burning before planting or during stand growth?
Line 102: For AP and K please generally state the type of method and analyzer used for measuring.
Line 133: What method/reference database was used for assigning taxonomy?
Line 134: How were data evaluated for normal distributions.
Line 136: this doesn't really evaluate the effect of stand age. It evaluates the difference among different stand ages in edaphic properties, microbial community structure, and enzyme activity. Please revise.
Line 146: I think you should note, in some cases these changes over time within soil depth were significant, in others they were not (see figure 2).
Line 163 (figure caption). It would be useful to write the three depths associated with D1-3 here.
Line 228: relative abundance in place of proportion...try to be consistent.
Line 232: Do not say "drove" . Your statistical analysis DOES NOT show this. Instead revise to something like "Although significant differences in relative abundance were observed among stand age plots...."
Line 284: THE (type - he)
Line 287: Not sure what is meant my "the better" . Does this have to do with C or N use efficiency? Or....? Please use a citation to support whatever your interpretation is here.
Line 286: MAY HAVE LED
Line 296: Again rather than saying stand age had effect (implying it drove changes), you revise to say "as soil properties changed significantly across stand age," which more appropriately reflects the statistical analyses employed.
Line 301: MAY BE influenced by
After line 302: You have to caveat the previous statement with something like: Although we find a correlation between microbial community diversity and enzyme activity, we note that this study does not reveal if this connection is simply a co-correlation to another soil factor, or a mechanism that may explain patters in enzyme activity. Further research is needed to address this question.
Line 322: may have promoted
Line 336: may be related....according to the correlations revealed in this study.
Line 347: Need to further qualify this statement. ,although further research is needed to understand if this correlation is related to some other variable or a direct relationship.
Line 355-357: Revise sentence, does not currently make sense. Possible revision:
Given that the stand age-driven changes in soil biotic and abiotic properties observed in this study are common to many plantations, our findings provide important insights into the impact of stand age on soil microbial communities and enzyme activities.
Line 357-359: Revise statement
In addition, we observed that changes in the richness and diversity of soil microbial communities across stand age correlated with enzyme activities. Given the connection between soil enzyme activity and microbial communities, our research suggests the need to further understand how soil microbial diversity may drive enzyme activity within forest management.
Author Response
Dear reviewer,
Thank you very much for your valuable comments on our manuscript. We have carefully revised the manuscript in accordance with the comments. We firmly believe that this will significantly improve the quality of our manuscript. Thank you again for the work you have done for our manuscript.
Kind regards
Authors
Point 1: Before line 43: 1) Start with general discussion of plant-microbe interaction and current knowledge, one type of suggest is something like:
Plant-microbe interactions are central to soil fertility and ecosystem function. In both natural and managed ecosystems research has demonstrated the strong selective effects of plants on soil microbial communities and how resultant soil microbial communities may directly impact plant communities and soil nutrient cycling (Van Der Heijden et al. 2008, Bueno de Mesquita 2017, Yuan et al. 2017)
Van Der Heijden, Marcel GA, Richard D. Bardgett, and Nico M. Van Straalen. "The unseen majority: soil microbes as drivers of plant diversity and productivity in terrestrial ecosystems." Ecology letters11.3 (2008): 296-310.
de Mesquita, Clifton P. Bueno, et al. "Plant colonization of moss-dominated soils in the alpine: Microbial and biogeochemical implications." Soil Biology and Biochemistry111 (2017): 135-142.
Yuan, Xia, et al. "Plant community and soil chemistry responses to long-term nitrogen inputs drive changes in alpine bacterial communities." Ecology 97.6 (2016): 1543-1554
Before line 51: You can connect these understandings more generally to your research, again cite relevant research in the field. Describe how increasing forest age may implicate microbial communities and the importance of these connections to ecosystem function. For example, add something like the following:
In general, studies of ecosystem succession have been central in demonstrating that forest ecosystems of increasing age can impact microbial community structure in variable ways over time. For example, there can be increasing selective effects of plant communities on microbial community structure and related function (Knelman et al. 2017, 2018 ). As microbial communities mediate biogeochemical cycles associated with nutrient and carbon cycling, in managed forest ecosystems the influence of increasing stand age on microbial communities may have strong effects on ecosystem functioning and soil fertility via soil enzyme activity (Holden et al. 2013, Smith et al 2015).
+Knelman, Joseph E., et al. "Interspecific Plant Interactions Reflected in Soil Bacterial Community Structure and Nitrogen Cycling in Primary Succession." Frontiers in microbiology 9 (2018): 128.
+Knelman, Joseph E., et al. "Rapid shifts in soil nutrients and decomposition enzyme activity in early succession following forest fire." Forests 8.9 (2017): 347.
+ Holden, Sandra R., Abraham Gutierrez, and Kathleen K. Treseder. "Changes in soil fungal communities, extracellular enzyme activities, and litter decomposition across a fire chronosequence in Alaskan boreal forests." Ecosystems 16.1 (2013): 34-46.
+Smith, A. Peyton, Erika Marín‐Spiotta, and Teri Balser. "Successional and seasonal variations in soil and litter microbial community structure and function during tropical postagricultural forest regeneration: a multiyear study." Global change biology 21.9 (2015): 3532-3547
Response 1: We adopted revised statement.
Plant-microbe interactions are central to soil fertility and ecosystem function. In both natural and managed ecosystems research has demonstrated the strong selective effects of plants on soil microbial communities and how resultant soil microbial communities may directly impact plant communities and soil nutrient cycling [1-3]. Soil enzymes and microorganisms are useful biotic indicators of soil fertility and ecosystem functions because they are the main drivers of soil nutrient and carbon cycling [4]. Catalase (CAT) is a specific enzyme for the metabolism of hydrogen peroxide [5]; sucrase (SC) plays an important role in decomposing sucrose and promoting the conversion of activated carbon [6]; β-glucosidase (BG) is mainly involved in the degradation of cellulose [4]; urease (UE) mainly promotes the mineralization of N [7]. Microorganisms play an important role in the production of enzymes, and the production of specific enzymes can be partly explained by the need of microorganisms for limiting nutrients [8].
In general, studies of ecosystem succession have been central in demonstrating that forest ecosystems of increasing age can impact microbial community structure in variable ways over time. For example, there can be increasing selective effects of plant communities on microbial community structure and related function [9, 10]. As microbial communities mediate biogeochemical cycles associated with nutrient and carbon cycling, in managed forest ecosystems the influence of increasing stand age on microbial communities may have strong effects on ecosystem functioning and soil fertility via soil enzyme activity [11, 12]. The results of a study on forest in the Mid-Atlantic US demonstrated that the concentrations of Mg, Ca, NO3 and pH in young forest were higher than those in old forest [13]. Zhang et al. [14] reported that soil nutrient contents and microbial biomass first gradually increased from<1a to 9a and then gradually decreased with the increase of stand age in Lycium barbarum L. plantation. Li et al. [15] found that certain correlations between soil microbial community diversity and stand age of seabuckthorn forest were obvious. Luo et al. [16] reported that soil enzyme activities, such as β-glucosidase, urease and protease, gradually increase with increasing stand age of Hippophae rhamnoides plantation. Therefore, determining the response of the microbial community and enzyme activity to changes in stand age can provide a theoretical basis for developing forest management strategies.
Point 2: line 44: What is meant by 'energy flow'. Do you just mean soil nutrient and carbon cycling? Please revise.
Response 2: We have replaced ‘soil nutrient cycling and energy flow’ with ‘soil nutrient and carbon cycling’.
Point 3: line 65: agreed UPON
Response 3: Nevertheless, the influence of stand age on soil enzyme activities and microbial diversity has not been agreed upon, and the effect of stand age on soil microbial community composition of Chinese fir plantations is still unclear.
Point 4: line 75/76: instead of "stand age will change" I would say "stand age will impact," because change more strongly suggests direct effects, while effects may be indirect.
Response 4: We have replaced ‘stand age will change’ with ‘stand age will impact’.
Point 5: Line 83/84: each plantation was given three plots WITHIN EACH BLOCK.
Response 5: The revision is as follows ‘each plantation was given three plots within each block’.
Point 6: Line 89: soil samples WERE stored at...
Response 6: The revision is as follows ‘Soil samples for analyses were immediately sieved (2 mm) and were stored at 4℃.’.
Point 7: Line 90: They were sieved again? In what way. Or if this is the same as the 2mm mentioned above please make sure the chronology of what you did is correctly communicated.
Response 7: The revision is as follow: ‘After transported to the laboratory, soil samples stored at -80℃ prior to microbial community analysis, or air dried and ground prior to chemical properties analysis.
Point 8: Line 91: How are these stands managed, if at all. Is there any thinning. Is there any fertilization or burning before planting or during stand growth?
Response 8: The following related statements have been added to the manuscript. The management measures for all plantations are consistent. All plantations were all burned before planting, and were not thinned and were not fertilized during stand growth.
Point 9: Line 102: For AP and K please generally state the type of method and analyzer used for measuring.
Response 9: The revision is as follows: AP and AK was extracted from 2.5 g of soil using 25 mL solution of a mixture of 0.03 mol/L NH4F and 0.025 mol/L HCL and 25 mL solution of 1 mol/L NH4Ac (pH=7.0), respectively. TP, TK, AP and AK contents were determined by ICAP (Spectro Analytical Instruments, Spectro Arcos ICP, Kleve, Germany).
Point 10: Line 133: What method/reference database was used for assigning taxonomy?
Response 10: The reference was added in the manuscript. The revision is as follows: After trimming, sequences were clustered into operational taxonomic units (OTUs) at 97% similarity level using Uclust [29].
Point 11: Line 134: How were data evaluated for normal distributions.
Response 11: We used Linear mixed model (LMM) to evaluate data for normality and homoscedasticity.
Linear mixed model (LMM) were used to test for variables affecting the variance in each enzyme and microbial diversity index in soil using physicochemical data as predictors, location and layer (nested within location) as random factors, and enzyme activities and microbial diversity indices as response variables [30]. To ensure normality and homoscedasticity, we applied a Box-Cox transformation with λ= 0.5 for dependent variables. To quantify unbiased measurements of variance expressed by fixed and fixed + random factors, respectively [31], we used conditional and marginal determination coefficients (R2c and R2m ) to express multilevel model goodness of fit. The R packages ‘lme4’ and ‘sjPlot’ were used to conduct the analyses.
Point 12: Line 136: this doesn't really evaluate the effect of stand age. It evaluates the difference among different stand ages in edaphic properties, microbial community structure, and enzyme activity. Please revise.
Response 12: ANOVA was conducted using SPSS (IBM Corp., New York City, USA) and used to evaluate the differences in the soil physicochemical properties, enzyme activities and abundance, diversity and composition of microbial community between different stand ages. Spearman’s correlation analyses were used to explore the relationships between soil enzyme activities and microbial community diversity of different stand ages.
Point 13:Line 146: I think you should note, in some cases these changes over time within soil depth were significant, in others they were not (see figure 2).
Response 13: The revision is as follows: Stand age affected soil pH and created an order of 49a> 32a> 25a> 18a> 12a> 3a> 6a.
Point 14:Line 163 (figure caption). It would be useful to write the three depths associated with D1-3 here.
Response 14: Figure 2. Soil physicochemical characteristics in the different age soils. Different lowercase letters and different uppercase letters represent significant differences between the same soil depth (D1: 0-10, D2: 10-20 or D3: 20-30 cm) of different stand ages and between different soil depths (D1: 0-10, D2: 10-20 and D3: 20-30 cm) of the same stand age at P< 0.05, respectively.
Point 15:Line 228: relative abundance in place of proportion...try to be consistent.
Response 15: The ‘proportion’ has been replaced with ‘relative abundance’. At the three soil depths, relative abundance of Proteobacteria all tended to first decrease when the stand age increased from 3a to 18a and then increase when stand age increased from 18a to 49a (Figure 4).
Point 16:Line 232: Do not say "drove" . Your statistical analysis DOES NOT show this. Instead revise to something like "Although significant differences in relative abundance were observed among stand age plots...."
Response 16: The revision is as follows: ‘Although significant differences in relative abundance of Actinobacteria, Chloroflexi, and Firmicutes were observed among stand age plots, these changes have no obvious regularity (Figure 4).’.
Point 17:Line 284: THE (type - he)
Response 17: However, the density of Chinese fir plantation decreased, the growth rate slowed down, the nutrient uptake decreased, and the litter stocks increased from 18a to 49a, which may have led to the increase of soil nutrient content.
Point 18:Line 287: Not sure what is meant my "the better". Does this have to do with C or N use efficiency? Or....? Please use a citation to support whatever your interpretation is here.
Response 18: The C: N ratio is an important indicator of soil C and N mineralization. The lower the C:N, the more favorable the release of N, and low rates of N release occur when C: N ratio exceeds 15 [33].
Point 19:Line 286: MAY HAVE LED
Response 19: However, the density of Chinese fir plantation decreased, the growth rate slowed down, the nutrient uptake decreased, and the litter stocks increased from 18a to 49a, which may have led to the increase of soil nutrient content.
Point 20:Line 296: Again rather than saying stand age had effect (implying it drove changes), you revise to say "as soil properties changed significantly across stand age," which more appropriately reflects the statistical analyses employed.
Response 20: Our results supported the first and partially supported the third hypothesis, as soil enzyme activities changed significantly across stand age, but stand age had not a positive or a negative effect on UE, SC and BG activities (Figure 3).
Point 21:Line 301: MAY BE influenced by
Response 21: The revision is as follows: ‘indicating that these enzyme activities in Chinese fir plantations were may be influenced by microorganisms.’
Point 22:After line 302: You have to caveat the previous statement with something like: Although we find a correlation between microbial community diversity and enzyme activity, we note that this study does not reveal if this connection is simply a co-correlation to another soil factor, or a mechanism that may explain patters in enzyme activity. Further research is needed to address this question.
Response 22: In our study, overall, the activities of UE, SC and BG in 12a, 18a and 25a were lower than 6a, 32a and 49a, which is in line with the trend of microbial diversity indices (Figure 3, Table 3), indicating that these enzyme activities in Chinese fir plantations were may be influenced by microorganisms. Although we find a correlation between microbial community diversity and enzyme activity, we note that this study does not reveal if this connection is simply a co-correlation to another soil factor, or a mechanism that may explain patters in enzyme activity. Further research is needed to address this question.
Point 23:Line 322: may have promoted
Response 23: As the increase of stand age from 25a to 49a, the increase in the composition and diversity of undergrowth vegetation, the amount of litter fall and nutrient contents may have promoted the increase of richness and diversity of microbial community.
Point 24:Line 336: may be related....according to the correlations revealed in this study.
Response 24: Given the strong correlation to organic substrates [45], the decrease in the relative abundance of Ascomycota from 6a to 18a or 25a and the increase of it from 25a to 49a may be related to litter fall stocks and SOM content according to the correlations revealed in this study.
Point 25: Line 347: Need to further qualify this statement,although further research is needed to understand if this correlation is related to some other variable or a direct relationship.
Response 25: Therefore, changes in soil microbial community composition caused by shifts in stand age may indirectly affect soil enzyme activities. However, further research is needed to understand whether this correlation is related to some other variable or a direct relationship.
Point 26: Line 355-357: Revise sentence, does not currently make sense. Possible revision:
Given that the stand age-driven changes in soil biotic and abiotic properties observed in this study are common to many plantations, our findings provide important insights into the impact of stand age on soil microbial communities and enzyme activities.
Response 26: We have adopted revised statement. Given that the stand age-driven changes in soil biotic and abiotic properties observed in this study are common to many plantations, our findings provide important insights into the impact of stand age on soil microbial communities and enzyme activities.
Point 27: Line 357-359: Revise statement
Response 27: We have adopted revised statement. In addition, we observed that changes in the richness and diversity of soil microbial communities across stand age correlated with enzyme activities. Given the connection between soil enzyme activity and microbial communities, our research suggests the need to further understand how soil microbial diversity may drive enzyme activity within forest management.

Reviewer 2 Report
The article "Contrasting effects of Chinese fir plantations of different stand ages on soil enzyme activities and microbial communities" is very interesting and rich in data. Nevertheless, the work requires a major revision.
The main concerns hampering its publication are as follows:
· The introduction, in particular from line 63 to line 77, looks like a discussion.
· Line 104: You talk about "litter fall", but, indeed, you do not analyze the litter that is falling but the litter standing on the soil. So, why do you not talk about "standing litter"?
· The applied statistical analysis exclusively highlights the differences between stands of different ages. Well! Given the large amount of data, however, the authors could perform more appropriate and integrated statistical tests, as, for example, the mixed model. In this case, the authors must make sure that the data have a normal distribution and that the variance is homogeneous. The reading of the following papers, in which the mixed model has been applied, can be suggested:
- Danise et. al 2018, Spectrophotometric methods for lignin and cellulose in forest soils as predictors for humic substances. European Journal of Soil Science, doi:10.1111/ejss.12678;
- Fioretto et al 2018, Discriminating between seasonal and chemical variation in extracellular enzyme activities within two Italian beech forests by means of multilevel models. Forests, doi:10.3390/f9040219
· Discussion: it is good that the authors cite the results of other authors, but the discussion of collected data should be highly increased and, above all, integrated. In this, a proper statistical test could help a lot.
· Conclusions: they must be improved and implemented.
Author Response
Dear reviewer,
Thank you very much for your valuable comments on our manuscript. We have carefully revised the manuscript in accordance with the comments. We firmly believe that this will significantly improve the quality of our manuscript. Thank you again for the work you have done for our manuscript.
Kind regards
Authors
Point 1: The introduction, in particular from line 63 to line 77, looks like a discussion.
Response 1: Although there are many studies on the soil properties of Chinese fir plantations, the results of soil enzyme activities and microbial changes driven by stand age are not consistent. The aim of our research is to comprehensively analyze the response mechanism of soil physical and chemical properties, enzyme activities and microbial community structure and composition to stand age change, so we conducted a simple comparative analysis of the findings of others in order to illustrate the necessity of our research.
Point 2: Line 104: You talk about "litter fall", but, indeed, you do not analyze the litter that is falling but the litter standing on the soil. So, why do you not talk about "standing litter"?
Response 2: Because we are focused on litter that has an impact on soil physicochemical properties and microbial communities, however, the litter standing has no effect on soil factors.
Point 3: The applied statistical analysis exclusively highlights the differences between stands of different ages. Well! Given the large amount of data, however, the authors could perform more appropriate and integrated statistical tests, as, for example, the mixed model. In this case, the authors must make sure that the data have a normal distribution and that the variance is homogeneous. The reading of the following papers, in which the mixed model has been applied, can be suggested:
- Danise et. al 2018, Spectrophotometric methods for lignin and cellulose in forest soils as predictors for humic substances. European Journal of Soil Science, doi:10.1111/ejss.12678;
- Fioretto et al 2018, Discriminating between seasonal and chemical variation in extracellular enzyme activities within two Italian beech forests by means of multilevel models. Forests, doi:10.3390/f9040219
Response 3: We adopted the revised suggestion and used the Linear mixed model (LMM) to evaluate data.
ANOVA was conducted using SPSS (IBM Corp., New York City, USA) and used to evaluate the differences in the soil physicochemical properties, enzyme activities and abundance, diversity and composition of microbial community between different stand ages. Spearman’s correlation analyses were used to explore the relationships between soil enzyme activities and microbial community diversity of different stand ages. Linear mixed model (LMM) were used to test for variables affecting the variance in each enzyme and microbial diversity index in soil using physicochemical data as predictors, location and layer (nested within location) as random factors, and enzyme activities and microbial diversity indices as response variables [30]. To ensure normality and homoscedasticity, we applied a Box-Cox transformation with λ= 0.5 for dependent variables. To quantify unbiased measurements of variance expressed by fixed and fixed + random factors, respectively [31], we used conditional and marginal determination coefficients (R2c and R2m ) to express multilevel model goodness of fit. The R packages ‘lme4’ and ‘sjPlot’ were used to conduct the analyses.
Point 4: Discussion: it is good that the authors cite the results of other authors, but the discussion of collected data should be highly increased and, above all, integrated. In this, a proper statistical test could help a lot.
Response 4: A linear mixed model (LMM) was used to test variables affecting the variance and microbial diversity index of each enzyme. The discussion has also been revised. Major revisions are as follows:
4.2. Shift in soil enzyme activities and composition and diversity of microbial communities
Our results supported the first and partially supported the third hypothesis, soil enzyme activities changed significantly across stand age, but stand age had not a positive or a negative effect on UE, SC and BG activities (Figure 3). Soil enzyme activity is closely related to microorganisms because soil enzymes are mainly secreted by microorganisms and plant roots [39, 40], which is also supported by our results of the correlations between enzyme activities and microbial diversity indices In our study, overall, the activities of UE, SC and BG in 12a, 18a and 25a were lower than 6a, 32a and 49a, which is in line with the trend of microbial diversity indices (Figure 3, Table 3), indicating that these enzyme activities in Chinese fir plantations were may be influenced by microorganisms. Although we find a correlation between microbial community diversity and enzyme activity, we note that this study does not reveal if this connection is simply a co-correlation to another soil factor, or a mechanism that may explain patters in enzyme activity. Further research is needed to address this question. Contrary to the third hypothesis, we found that the stand age of Chinese fir plantation had a positive influence on CAT activity (Figure 3a). However, Shu [18] reported that the CAT activity at 0-250px depth created an order of 8a> 11a> 15a, while it at 10-20 cm and 20-750px depths created an order of 8a> 15a> 11a. Gu et al. [41] reported that the decrease in stand density can stimulate CAT activity, which may partly explain the increase in CAT activity. Vertical changes in soil enzyme activities were found in many studies [42, 43].
4.3. Effects of soil physicochemical properties on enzyme activities and microbial diversity
Previous studies reported that soil chemical properties such C, N, P and K had significant effects on microbial community [54, 55] Many studies suggested that soil enzyme activities and microorganisms are influenced by the soil pH [52, 56]. In this study, however, TN and AP contents were primary influencing factors and more important than soil pH in terms of soil enzyme activities and microorganisms responses to stand age change drivers. The LMM results showed that enzyme activities and microbial community diversities were also affected by random factors, indicating that the use of the LMM is more conducive to determining predictors of enzyme activities and microbial communities. We found that the correlation of the richness and diversity of soil microbial community with soil enzyme activities was significant (Table 5). Therefore, changes in soil microbial community composition caused by shifts in stand age may indirectly affect soil enzyme activities. However, further research is needed to understand whether this correlation is related to some other variable or a direct relationship.
Point 5: Conclusions: they must be improved and implemented.
Response 5: The revision is as follows:
Our study demonstrated that the stand age of Chinese fir plantation can drive changes in soil physicochemical properties, enzyme activities and richness and diversity of microbial community at 0-30 cm depth. Low stand age (3a and 6a) and high stand age (32a and 49a) showed high nutrient contents, the activities of UE, SC and BG, and the richness and diversity of microbial community compared to middle stand age (12a, 18a and 25a). Given that the stand age-driven changes in soil biotic and abiotic properties observed in this study are common to many plantations, our findings provide important insights into the impact of stand age on soil microbial communities and enzyme activities. In addition, we observed that changes in the richness and diversity of soil microbial communities across stand age correlated with enzyme activities. Given the connection between soil enzyme activity and microbial communities, our research suggests the need to further understand how soil microbial diversity may drive enzyme activity within forest management.

Round 2
Reviewer 2 Report
The article "Contrasting effects of Chinese fir plantations of different stand ages on soil enzyme activities and microbial communities" has been well revised by the Authors.
1) Soil pH was measured using a suspension of soil and water in a ratio of 1:5. Generally, soil and water are in a ratio of 1:2.5. Why the authors did use that ratio? Is it documented? Accordingly, a reference should be inserted.
2) The authors keep writing about “litterfall”. Such term is not suitable. Litterfall refers to the litter that is falling or has just fallen to the ground. In support of this terminology there is a large literature and, as an example, here is a few works:
- H. Zhang, W. Yuan, W.D. ShuguangLiu. 2014 - Seasonal patterns of litterfall in forest ecosystem worldwide
Ecological Complexity, vol. 20: 240-247.
- L. Cizungu et al. 2014 - Litterfall and leaf litter decomposition in a central African tropical mountain forest and Eucalyptus plantation
Forest Ecology and Management, 326 : 109–116
- J.A. Blanco, 2018 - Managing Forest Soils for Carbon Sequestration: Insights From Modeling Forests Around the Globe. Effects on Organic Carbon, Nitrogen Dynamics, and Greenhouse Gas Emissions.,
In: Soil Management and Climate Change (María Ángeles Muñoz and Raúl Zornoza Editors)
- J. Barlow, et al. 2007 - Litter fall and decomposition in primary, secondary and plantation forests in the Brazilian Amazon
Forest Ecology and Management 247(1):91-97
- B. Berg & R. Laskowski , 2005- Litter Fall.
Advances in Ecoloogical Research, vol.38: 19-71
Just to name a few!!!
If the authors do not like the suggested term of "standing litter" they can write about “organic horizon of the soil”. Certainly, the answer they gave in the rebuttal letter to the reviewer two (point 2) makes no sense and I do not understand what do they mean when they say that "the litter standing has no effect on soil factors". Do the authors have a clear understanding of what we are talking about?
Author Response
Dear review,
Thank you very much again for your valuable comments on our manuscript. We apologize for the lack of proper understanding of the comment of replacing litter fall with standing litter. We have revised the manuscript based on the comments. Thank you again for your work on our manuscript.
Kind regards
Authors
Point 1: Soil pH was measured using a suspension of soil and water in a ratio of 1:5. Generally, soil and water are in a ratio of 1:2.5. Why the authors did use that ratio? Is it documented? Accordingly, a reference should be inserted.
Response 1: The reference was inserted in the manuscript.
Soil pH was measured using a suspension of air-dried soil and distilled water (in a ratio of 1:5) [24].
[24] Nguyen, L.T.T.; Osanai, Y.; Lai, K.; Anderson, I.; Bange, M.; Tissue, D. Responses of the soil microbial community to nitrogen fertilizer regimes and historical exposure to extreme weather events: flooding or prolonged-drought. Soil Biol. Biochem. 2018, 118, 227-236.
Point 2: The authors keep writing about “litterfall”. Such term is not suitable. Litterfall refers to the litter that is falling or has just fallen to the ground. In support of this terminology there is a large literature and, as an example, here is a few works:
- H. Zhang, W. Yuan, W.D. ShuguangLiu. 2014 - Seasonal patterns of litterfall in forest ecosystem worldwide
Ecological Complexity, vol. 20: 240-247.
- L. Cizungu et al. 2014 - Litterfall and leaf litter decomposition in a central African tropical mountain forest and Eucalyptus plantation
Forest Ecology and Management, 326 : 109–116
- J.A. Blanco, 2018 - Managing Forest Soils for Carbon Sequestration: Insights From Modeling Forests Around the Globe. Effects on Organic Carbon, Nitrogen Dynamics, and Greenhouse Gas Emissions.,
In: Soil Management and Climate Change (María Ángeles Muñoz and Raúl Zornoza Editors)
- J. Barlow, et al. 2007 - Litter fall and decomposition in primary, secondary and plantation forests in the Brazilian Amazon
Forest Ecology and Management 247(1):91-97
- B. Berg & R. Laskowski , 2005- Litter Fall.
Advances in Ecoloogical Research, vol.38: 19-71
Just to name a few!!!
If the authors do not like the suggested term of "standing litter" they can write about “organic horizon of the soil”. Certainly, the answer they gave in the rebuttal letter to the reviewer two (point 2) makes no sense and I do not understand what do they mean when they say that "the litter standing has no effect on soil factors". Do the authors have a clear understanding of what we are talking about?
Response 2: We have replaced the litter fall with standing litter. Sorry, we did not understand the reviewer's comment correctly before. I am very grateful to the reviewers for correcting our misrepresentation.
2.3. Standing litter analyses
3.2. Standing litter stocks
4.1. Shift in soil pyhsicochemical properties and standing litter stocks
